# Transgenic shRNA pigs reduce susceptibility to foot and mouth disease virus infection

**Shengwei Hu[1,2†], Jun Qiao[1†], Qiang Fu[1†], Chuangfu Chen[1]\*, Wei Ni[2]\*, Sai Wujiafu[1], Shiwei Ma[1], Hui Zhang[1], Jingliang Sheng[1], Pengyan Wang[1], Dawei Wang[1], Jiong Huang[3], Lijuan Cao[1], Hongsheng Ouyang[4]\***

[1]College of Life Sciences, Shihezi University, Shihezi, China; [2]College of Animal Sciences, Shihezi University, Shihezi, China; [3]Institute of Veterinary Medicine, Xinjiang Academy of Animal Science, Urumqi, China; [4]College of Animal Science and Veterinary Medicine, Jilin University, Changchun, China

**Abstract** Foot-and-mouth disease virus (FMDV) is an economically devastating viral disease leading to a substantial loss to the swine industry worldwide. A novel alternative strategy is to develop pigs that are genetically resistant to infection. Here, we produce transgenic (TG) pigs that constitutively expressed FMDV-specific short interfering RNA (siRNA) derived from small hairpin RNA (shRNA). In vitro challenge of TG fibroblasts showed the shRNA suppressed viral growth. TG and non-TG pigs were challenged by intramuscular injection with 100 $LD_{50}$ of FMDV. High fever, severe clinical signs of foot-and-mouth disease and typical histopathological changes were observed in all of the non-TG pigs but in none of the high-siRNA pigs. Our results show that TG shRNA can provide a viable tool for producing animals with enhanced resistance to FMDV.

**\*For correspondence:**
chencf1962@yahoo.com (CC);
niweiwonderful@sina.com (WN);
28021883@qq.com (HO)

[†]These authors contributed equally to this work

**Competing interests:** The authors declare that no competing interests exist.

**Reviewing editor**: Stephen P Goff, Howard Hughes Medical Institute, Columbia University, United States

## Introduction

Foot-and-mouth disease (FMD) is an acute and highly contagious disease of cloven-hoofed animals, including cattle, pigs, sheep and goats and more than 70 wildlife species, and is devastating especially in young animals (*Grubman and Baxt, 2004*). The etiological agent of FMD is foot-and-mouth disease virus (FMDV), which belongs to the genus aphthovirus of the family Picornaviridae (*Bachrach, 1968*). Some control strategies including eradication, vaccination, selective test and slaughter have been widely used for preventing FMDV infection (*Leforban, 1999*; *Barnett and Carabin, 2002*), but diseases caused by FMDV remain prevalent in pigs and cattle all over the world owing to the absence of reciprocal protection among several FMDV serotypes (*Haydon et al., 2001*).

RNA interference (RNAi) is a post-transcriptional process initiated by double-stranded RNA (dsRNA) homologous to a target gene sequence (*Meister and Tuschl, 2004*). Specific gene silencing can also be triggered in mammalian cells by using synthetic short interfering RNA (siRNA), and plasmid or virus-mediated short hairpin RNA (shRNA) (*Elbashir et al., 2001*; *Hammond et al., 2001*; *Paul et al., 2002*; *Michel et al., 2005*; *Kim and Rossi, 2007*). The shRNA was proposed as a therapy for suppressing the infection of FMDV in vitro and in vivo (*Chen et al., 2006*; *Kim et al., 2008*). Recently, we extended that finding by producing genetically engineered mice integrating shRNA targeting FMDV (*Pengyan et al., 2008*, *2010*). The majority of transgenic (TG) mice infected with FMDV were resistant to infection and showed only slightly abnormal pathology compared with controls. Now, we report that TG pigs expressing siRNA against FMDV are resistant to viral challenge.

**eLife digest** Foot-and-mouth disease regularly causes serious outbreaks in livestock. The virus that causes the disease can infect cattle, pigs, sheep, goats, and many species of wild animals; the disease is also highly contagious and spreads very quickly and easily. To control the spread of foot-and-mouth disease, farmers must often kill entire herds of animals that have been exposed. Wild animals that can spread the virus may also be killed in an effort to stop the spread of the disease.

Vaccines that protect against foot-and-mouth disease are available and are often used to help prevent the spread of the disease. However, once an outbreak of foot-and-mouth disease begins it may be too late for vaccines to stop its spread. This is because the vaccines can take about a week to provide protection, and by that time an exposed animal may already be very ill.

Previous work conducted in 2010 reported that mice could be genetically engineered to produce short stretches of RNA molecules that can switch off genes from the foot-and-mouth disease virus. Compared with normal mice infected with the foot-and-mouth disease virus, the genetically engineered mice showed little sign of the disease in their bodies. Now, Hu, Qiao, Fu et al.—including some of the researchers involved in the 2010 work—have genetically engineered pigs in the same way. The experiments show that when cells from these pigs are exposed to the foot-and-mouth disease virus in the laboratory, the virus grows much less than normal.

Next, Hu, Qiao, Fu et al. injected genetically engineered pigs and non-genetically engineered pigs with the virus. All of the normal pigs developed severe symptoms very quickly, including the disease's characteristic mouth and foot sores. Additionally, examinations of these pigs' cells showed signs of the disease. But the genetically engineered pigs did not become seriously ill and their cells showed little sign of the disease. Some of the genetically engineered pigs developed a few sores but these sores appeared much later than normal. So far, the results suggest that it may be possible to develop pigs that are resistant to foot-and-mouth disease. Hu, Qiao, Fu et al. will next determine whether or not the genetically engineered pigs can pass the foot-and-mouth virus on to other pigs and livestock.

## Results and discussion

We constructed a total of 10 shRNA expression vectors (*Figure 1A*) targeting viral structural protein VP1 of FMDV type O and determined the efficacy of shRNAs for inhibiting FMDV replication in BHK cells by real-time RT-PCR. The V3 shRNA reduced the expression of viral RNA by 96.8% compared to scrambled control (*Figure 1B*). The V3 shRNA expression vector was used to generate TG pigs by somatic cell nuclear transfer. A total of 42 pigs were born alive, eight of which survived at least 6 months. TG pigs contained the stably integrated transgene as evidenced by PCR (*Figure 2A*) and green fluorescent protein (GFP) expression (*Figure 2B* and *Figure 2—figure supplement 1*). The copy numbers of transgene were measured by real-time PCR. The copy numbers of the inserted vector were calculated to be 3–11 (*Figure 2C*). Expression of siRNA in fibroblast cells isolated from TG pigs was examined by custom TaqMan small RNA assays (*Figure 2D*). The siRNA expression in TG 11, 19, 69 and 101 was 10–30-fold that from TG 24, 45, 49 and 78. After necropsy of TG 69 and 101, expression of siRNA was detected in various tissues, including heart, lung, spleen, liver, kidney and muscle, although the siRNA levels were diverse among different tissues (*Figure 2E*).

Next, we tested shRNA transgene resistance to FMDV infection in fibroblast cells isolated from high-siRNA TG (11, 19, 69 and 101), low-siRNA TG (24, 45, 49 and 78) and non-TG pigs. Compared to non-TG cells, viral RNA expression was reduced by 30-fold in high-siRNA TG and 12-fold in low-siRNA TG cells at 36 hr after virus challenge (*Figure 3A*). Inhibition of FMDV infection was a positive correlation with siRNA expression in fibroblast cells (*Figure 2B* and *Figure 3A*). Moreover, TG fibroblasts visibly reduced FMDV-induced cytopathogenic effects as compared with non-TG fibroblasts (*Figure 3B*).

The resistance of TG pigs to FMDV infection was further tested by intramuscular injection of O serotypes of FMDV. The challenged animals included high-siRNA TG (11 and 19), low-siRNA TG (24, 49 and 78) and non-TG pigs (n=5). Prior to the day of infection, no animal tested was positive for FMDV. After FMDV challenge, all non-TG pigs developed high fever within 72 hr of challenge and severe clinical signs of FMD, the appearance of vesicles on the feet and nose (*Figure 4A* and *Figure 4—figure supplement 1*). All non-TG pigs became deteriorated and the lesion score reached

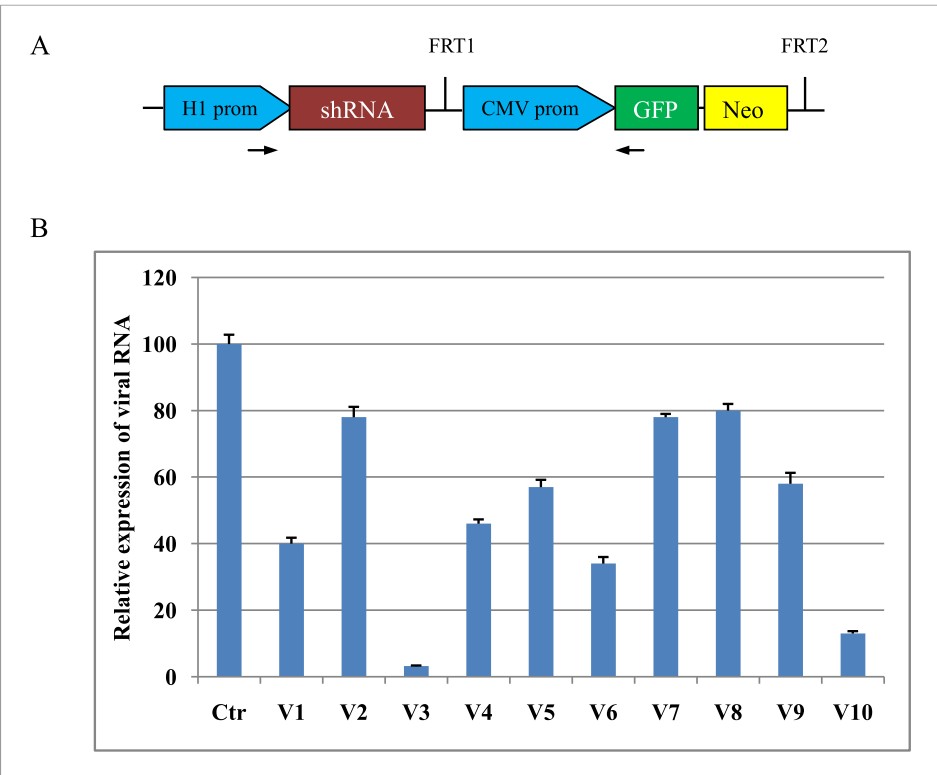

**Figure 1**. Design and screening of shRNA expression vector. (**A**) Schematic diagram of shRNA expression vector (pXL-EGFP-NEO) used. This vector includes a mouse H1 RNase promoter driving ubiquitous expression of shRNA and a cytomegalovirus-immediate early (CMV) promoter driving GFP and neomycin fusion expression. The arrows denote the PCR primers spanning H1 promoter, shRNA and GFP elements used to identify transgene integration in cloned pigs. (**B**) Relative expression of viral RNA in shRNA-transfected BHK cells. Data are means of three replicates ±SD.

24 at 5 d after challenge (*Figure 4A* and *Figure 4—figure supplement 1*). Some smaller vesicles in low-siRNA TG (24, 49 and 78) pigs were also observed until 7 d after challenge, as shown in *Figure 4A*. However, the body temperature of high-siRNA TG (11 and 19) pigs remained normal throughout the experiment (*Figure 4A* and *Figure 4—figure supplement 1*). TG pigs 11 and 19 developed one small vesicle at day 9 of challenge, but TG pig 11 recovered soon on the next day (*Figure 4A*). We subsequently quantified the viral genome RNA in the serum of the infected animals. Consistent with the clinical signs data, the viral load in the serum of the high-siRNA TG and low-siRNA TG pigs was lower than that in the non-TG pigs (*Figure 4B*). The viral RNA expression in serum was 42-fold lower in the high-siRNA TG group than that in the non-TG pigs at day 10 of challenge.

All animals were killed on the 10th day post-infection, and major tissues including lesions were collected for levels of virus RNA and histopathology analysis. Viral RNA was not detected in the heart, lung, spleen, liver, kidney and muscle in the challenged TG pigs, but viral RNA still maintained high levels of expression in lymph and lesions in the non-TG pigs, except heart, lung, spleen and liver (*Figure 4C*). No viral RNA in the non-TG heart, lung, spleen and liver showed clearance of viral RNA from the tissues, consistent with prior findings (*Zhang and Alexandersen, 2004*; *Chen et al., 2006*). Furthermore, lesions, as a potential source of virus transmission by aerosol, were well known to be the predominant tissue site of FMDV infection and amplification (*Zhang and Bashiruddin, 2009*; *Dillon, 2011*). Levels of viral RNA in foot lesions of TG pigs were much lower than those in the non-TG pigs (*Figure 4C*), suggesting an encouraging result for blocking transmission.

Haematoxilin/eosin (HE) staining of major tissues revealed that non-TG pigs had severe abnormal pathology compared to TG pigs. In particular, non-TG pigs showed multifocal necrosis in the liver, and interstitial pneumonia and severe congestion in the lung (*Figure 4D*). None of the TG pigs showed typical histopathological changes, except one case of interstitial pneumonia.

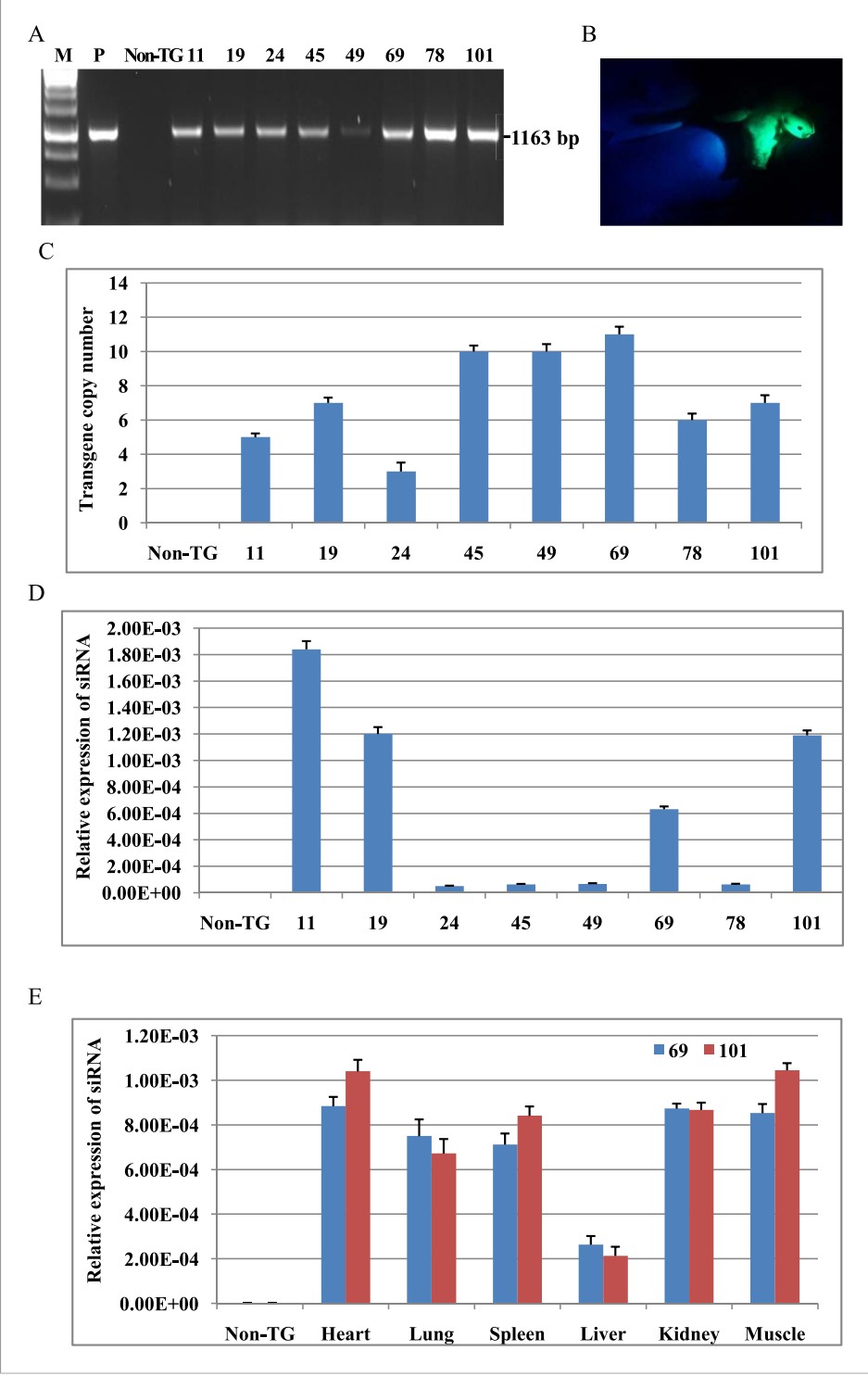

**Figure 2**. Analysis of shRNA transgene in cloned pigs. (**A**) PCR for detecting shRNA expression cassette. PCR products spanned H1 promoter, shRNA and GFP cassette. P: plasmid as positive control. Non-TG: non-TG pig as negative control. 11, 19, 24, 45, 49, 69, 78 and 101: cloned pigs. (**B**) EGFP fluorescence of transgenic pigs. (**C**) The copy numbers of transgene were determined by real-time PCR. (**D**) Analysis of siRNA expression in fibroblast cells of all transgenic pigs. (**E**) Analysis of siRNA expression in various tissues of TG 69 and 101. Data are presented as means of three replicates ±SD.

The following figure supplement is available for figure 2:

**Figure supplement 1**. Pictures of transgenic pigs and EGFP expression in the fibroblast cells.

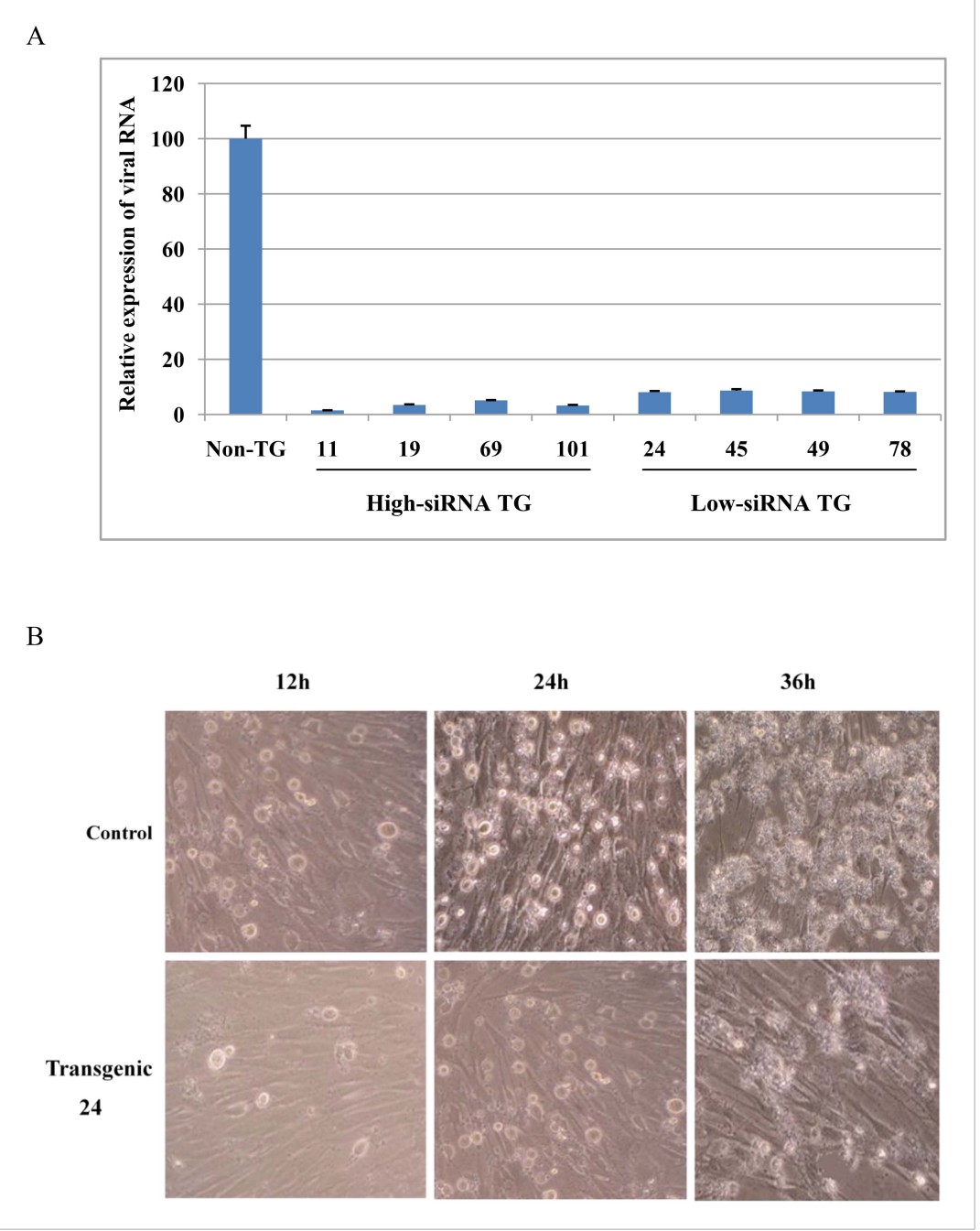

**Figure 3**. shRNA transgene resistance to FMDV infection in fibroblast cells of transgenic pigs. (**A**) Relative expression of viral RNA in fibroblast cells after FMDV infection. Data are presented as means ±SD. (**B**) Fibroblast cells were observed for development of cytopathogenic effect by bright-field microscopy at 12, 24 and 36 hr post-infection.

Recently, *Lyall et al. (2011)* in *Science* reported that onward transmission of avian influenza in TG shRNA chickens was prevented, although the TG chickens succumbed to the initial direct challenge, leading to a strategy for controlling avian influenza outbreaks. Our results showed that the TG pigs exhibited a marked resistence to FMDV infection after direct challenge. As encouraging as these results are, an onward transmission experiment will be performed in the future when producing enough high-siRNA TG pigs.

The most important threat caused by FMDV is the high speed of viral replication, short incubation time, and high contagiousness. Although protective immune responses from vaccination against FMDV

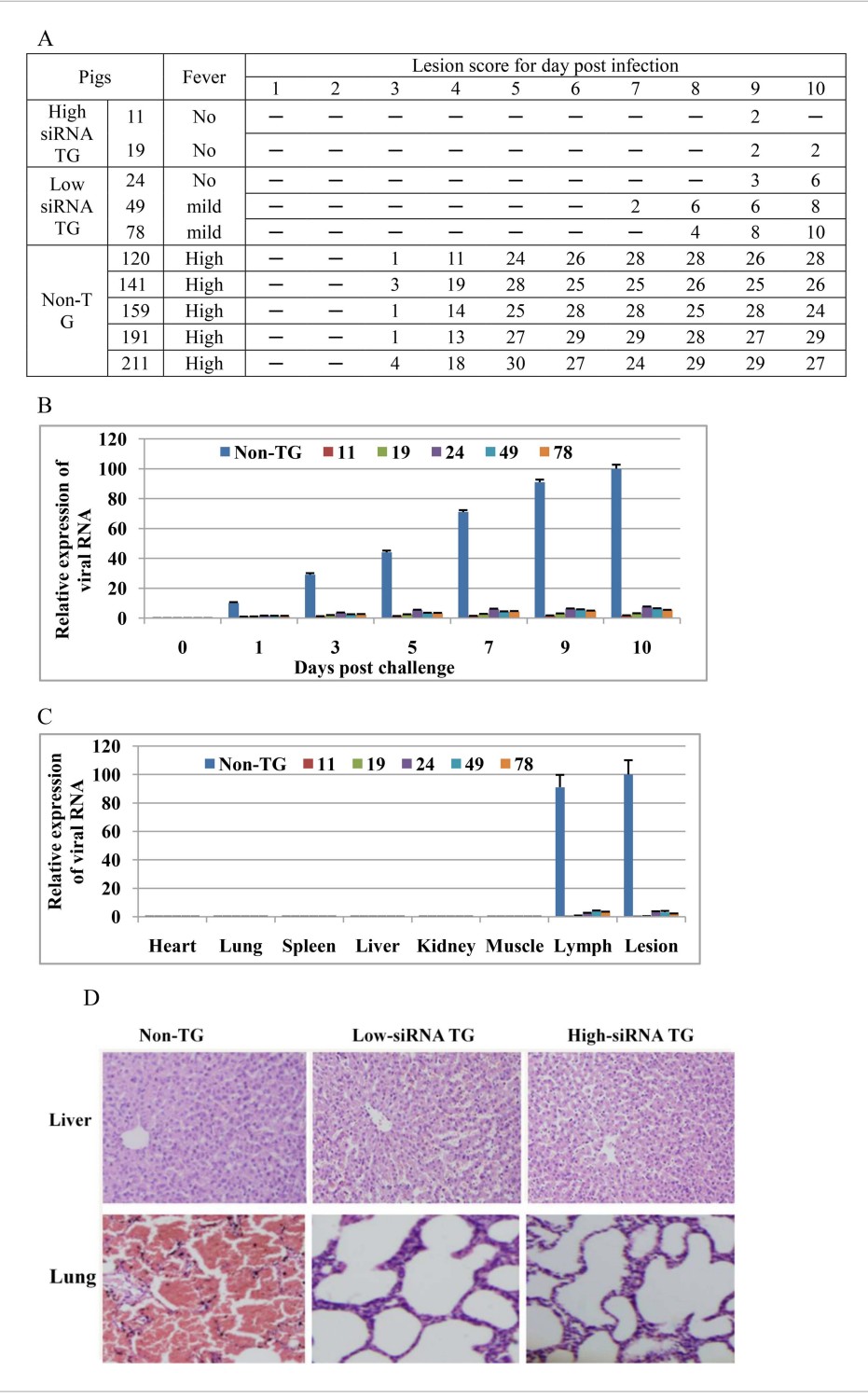

**Figure 4**. Transgenic shRNA pigs resisted FMDV infection. (**A**) Clinical sign of TG and non-TG pigs challenged with O serotypes of FMDV. Body temperature was detected every day after infection. Body temperature 38–39.5°C (no fever); body temperature up to 39.5–40°C (mild fever); body temperature over 40°C (high fever). Lesion score based on the appearance of vesicles on the feet and nose (see 'Materials and methods'). None of vesicles on the feet and nose (–). (**B**) Relative expression of viral RNA in serum of the infected animals. Data are presented as means ±SD. (**C**) Relative expression of viral RNA in various tissues of the infected animals. Data are presented as means ±SD. (**D**) HE

*Figure 4. continued on next page*

*Figure 4. Continued*

staining of tissue sections from non-TG and TG pigs. HE staining revealed that there was hepatic multifocal necrosis in non-TG pigs and interstitial pneumonia and severe congestion in the lung of non-TG pigs.

The following figure supplement is available for figure 4:

**Figure supplement 1**. Body temperature curve of all infected pigs.

can be efficacious, the rapidity of virus replication and spread can outpace the development of immune defenses and overrun the immune system (*Summerfield et al., 2009*). Current FMDV vaccines do not induce a protective response until 7 d post-vaccination (*Barnett et al., 2002*; *Doel, 2003*). FMD signs in high-siRNA TG pigs in our study were delayed for at least 8 d after FMDV infection (*Figure 4A*). siRNA expressed in TG pigs can also play a role as co-agent to induce rapid resistance before routine vaccination can evoke protective immunity. TG siRNA pigs immunized with current vaccines may achieve complete protection for an FMDV outbreak, which provides a novel strategy for preventing FMD in a disease-free country.

The shRNA-based transgene strategy has substantial benefits over vaccination by offering potential sub-serotype protection when using multiple-shRNA expression systems targeting different viruses (*Cong et al., 2010*). Our findings demonstrate that RNAi technology combining animal cloning offers the possibility to produce TG animal with improved resistance to viral infection.

## Materials and methods

### Design of shRNAs and plasmids

Conserved sequences such as the siRNA target site had been reported as an alternative strategy preventing the escape mutants of virus (*Dave and Pomerantz, 2004*). Conserved target sequences were selected from the viral structural protein VP1 gene by sequence alignment of O, A and Asia 1 serotypes of FMDV. The shRNA was designed by using the Ambion website tool (http://www.ambion.com/techlib/misc/siRNA_finder.html). These shRNA sequences are summarized in *Supplementary file 1*. Oligonucleotides were annealed and cloned into the pXL-EGFP-NEO to generate a series of shRNA expression plasmids (*Figure 1A*).

### Screening of shRNAs for inhibiting FMDV infections

BHK cells were seeded in 24-well plates (CoStar, Cambridge, MA) the day before transfection to achieve 90% confluency. The cells were transfected with 2.5 µg shRNA expression plasmids using Lipofectamine 2000 (Invitrogen, Carlsbad, CA). After 12 hr of transfection, the transfection complex was removed and the cells were washed twice with DMEM. The transfected cells in per well plates were then infected with 100 $TCID_{50}$ of FMDV (OS/99 strain). After 1 hr of adsorption, the inoculum was removed and the cells were washed twice with DMEM. The infection then proceeded in DMEM supplemented with 2% fetal bovine serum. Virus samples were collected at designated time points and frozen at −80°C until assessment of viral RNA.

### Real-time RT-PCR for viral RNA

Viral RNA was isolated using Trizol (Invitrogen) according to the manufacturer's instructions. From purified RNA, complementary DNA was synthesized using random hexamer primers and was quantified by spectrophotometer at 260 nm. Real-time PCR was carried out using SYBR Green (TaKaRa Biotech, Dalian, China) following the manufacturer's protocol. The following primers were used for FMDV VP1 gene amplification (VP1-F: 5′-TCA AGC CAA AGG AAC AAGT-3′; VP1-R: 5′-TAG ACG GTC GCT AAG ACAC-3′). GAPDH served as an internal control. The ΔΔCt method was used for relative quantification (*Livak and Schmittgen, 2001*).

### Generation of TG pigs

Pig primary fibroblasts were isolated as previously described (*Fan et al., 2013*). The fibroblast cells were transfected with linearized shRNA expression vectors, and then were split into 12-well

plates at an appropriate dilution (2000 cells/well) for G418 selection (400 µg/ml; Promega, Madison, WI) (*Cong et al., 2010*). G418-resistant and GFP-positive colonies derived from individual cells were obtained after 14 d of culture. The positive cells were used for somatic cell nuclear transfer as described previously (*Li et al., 2009*; *Ni et al., 2014*). Approximately 200–300 embryos were transferred into each surrogate pig. Cloned pigs were delivered by natural birth at full term.

## PCR analysis

Genomic DNA was isolated from ear biopsies of cloned pigs using the TIANamp genomic DNA kit (Tiangen Biotech, China). Transgene integration was identified by PCR assays. PCR was performed on 400 ng of genomic DNA using specific primers (H1-F: TGT CGC TAT GTG TTC TGGG; GFP-R: TGT CTT GTA GTT CCC GTC ATC) for amplifying shRNA and GFP expression cassette. PCR reaction consisted of 95°C for 4 min; 30 cycles at 95°C for 30 s, 57°C for 30 s and 72°C for 50 s; an extension at 72°C for 5 min. PCR products were analyzed by 1% gel electrophoresis.

## Determination of transgene copy number

The copy numbers of transgenes were determined by real-time PCR as previously described (*Kong et al., 2009*). Briefly, a standard curve was produced with series of standard samples containing 1, 2, 4, 8, 10, 12, 16 copies of the shRNA gene, respectively, by mixing the wild-type genome of pig with shRNA expression vector. The absolute quantitative standard curve was drawn by plotting $\ddot{A}Ct$ ($\ddot{A}Ct=Ct_{shRNA}-Ct_{TFRC}$) against the log of shRNA gene copies of corresponding standard samples.

## siRNA expression analysis

siRNA expression in TG pigs was determined by using TaqMan small RNA assays (Applied Biosystems, Foster City, CA) (*Chen et al., 2005*). Small RNAs were isolated by using the mirVana miRNA isolation Kit (Ambion, Austin, TX). Real-time RT-PCR was performed according to the manufacturer's instructions. Endogenous U6 was used as a RNA quality and loading control. The shRNA expression was normalized to the expression of U6 using the $2^{-\Delta\Delta Ct}$ method (Ct of shRNA–Ct of U6) (*Livak and Schmittgen, 2001*).

## In vitro challenge of TG fibroblasts

TG fibroblasts were isolated from ear biopsies of cloned pigs as previously described (*Li et al., 2014*). The cells cultured in 96-well plates were inoculated with 100 $TCID_{50}$ of O serotypes of FMDV (OS/99 strain). After 1 hr absorption, the inoculum was removed and the infection then proceeded in DMEM supplemented with 2% fetal bovine serum. The infected cells were observed for cytopathic effects at 12, 24 and 36 hr post-challenge. Virus samples were collected at 36 hr post-challenge. Relative expression of viral RNA was evaluated by real-time RT-PCR as described above.

## Viral challenge assay in TG pigs

All experiments involving animals were conducted under the protocol (SU-ACUC-12031) approved by the Animal Care and Use Committee of Shihezi University. Viral challenge was performed with O serotypes of FMDV (OS/99 strain). The challenged pigs (10–13 months of age) included high-siRNA TG (11 and 19), low-siRNA TG (24, 49 and 78) and non-TG controls (n=5). Before virus challenge, all animals were confirmed as negative for FMDV infection. All animals were housed in one room and challenged by intramuscular injection with 100 $LD_{50}$ in 1 ml of phosphate-buffered saline (PBS) in the neck area.

After challenge, animals were examined daily for clinical signs of FMD, including body temperature and the appearance of vesicles on the nose, mouth and feet. Body temperature remaining at 38–39.5°C was defined as no fever, body temperature up to 39.5–40°C was defined as mild fever, and body temperature over 40°C was defined as high fever. The lesion score was calculated by determining the number and size of vesicles on the nose, mouth and feet of each animal; 1 cm of each vesicle was recorded as 1, 2 cm were recorded as 2, and other larger vesicles were recorded as 3 (if on the nose or mouth) or 6 (if on the feet). The observations were terminated on day 10 post-challenge when the animals were killed.

## Quantification of serum viral RNA

Blood samples were collected at days 0, 1, 3, 5, 7, 9 and 10 after challenge. Total RNA was extracted from blood and subjected to real-time RT-PCR as described above.

## Histopathology analysis

All animals were killed on the 10th day post-infection, and major tissues were fixed in formalin for 10 hr followed by routine paraffin sectioning and HE staining. Histopathological changes were observed under microscope.

## Acknowledgements

This work was supported by grants from International S&T Cooperation Program of China (grant 2013DFR30970), Natural Science Foundation of China (grants 31101816 and 31360615) and Projects of Cultivating New Varieties by Transgenic Technology (2009ZX08005-003B).

## Additional information

### Funding

| Funder | Grant reference | Author |
| --- | --- | --- |
| National Natural Science Foundation of China (NSFC) | 31101816 | Wei Ni |

The funder had no role in study design, data collection and interpretation, or the decision to submit the work for publication.

### Author contributions

SH, JQ, CC, WN, Conception and design, Acquisition of data, Analysis and interpretation of data, Drafting or revising the article, Contributed unpublished essential data or reagents; QF, JH, LC, Conception and design, Acquisition of data, Analysis and interpretation of data, Drafting or revising the article; SW, SM, HZ, PW, DW, HO, Conception and design, Acquisition of data, Analysis and interpretation of data, Contributed unpublished essential data or reagents; JS, Conception and design, Analysis and interpretation of data, Contributed unpublished essential data or reagents

### Ethics

Animal experimentation: All experiments involving animals were conducted under the protocol approved by the Animal Care and Use Committee of Shihezi University (SU-ACUC-12031).

## Additional files

### Supplementary file

• Supplementary file 1. Target sequences of shRNA used in this study.

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
