## [Decision Letter]

Thank you for sending your work entitled “Transgenic shRNA pigs resist foot and mouth disease virus infection” for consideration at *eLife*. Your article has been favorably evaluated by Charles Sawyers (Senior editor) and two reviewers, one of whom is a member of our Board of Reviewing Editors.

The Reviewing editor and the other reviewer discussed their comments before we reached this decision, and the Reviewing editor has assembled the following comments to help you prepare a revised submission.

We reviewers were in near perfect agreement that your paper is exciting but had some deficiencies that need to be addressed.

Most important is the need for further analysis of the status of the virus in the experimental and control animals. These include the expression of the siRNAs in tissues; DNA copy number of the transgene; the virus titers/viremia from plasma and perhaps in nasal secretions; and issues of strains of virus. Please try to address as many of these issues as possible in a revised draft. We appreciate that there are severe technical limitations in working with large animal models, but more data on the course of the disease is important for the paper.

Reviewer #1:

This short paper reports the generation of a set of transgenic pig lines that express an shRNA targeting the FMDV RNA, and documents the resistance of some of these lines to pathology upon challenge with high doses of FMDV. Both high-level shRNA expressors and low-level expressors were obtained and showed resistance. This is an exciting development with important implications for the management of this troublesome and economically significant virus. Enormous efforts have been invested in the production of vaccines, with only limited success. This approach offers significant potential benefits to the livestock industry.

The paper lacks much detail about the gene expression and virus inhibition. Some of the information that one would like to see:

1) How many copies of the transgene are present in each of the eight lines? What tissues (beyond fibroblasts) express the transgene, and at what levels? Is it not surprising that both the high-expressing and low-expressing lines show substantial blocks to virus replication?

2) We are given body temperature and vesicle count, but little more about the course of disease (or lack thereof). We have serum levels of virus RNA, but where does virus replicate most extensively (if at all)? Are pathology sections completely free of viral RNA or just indicating lower levels of virus RNA? Is virus shed? (This readout should be very sensitive.) When is it cleared? Does virus persist (for as long as the animals are maintained?) Given the importance of blocking transmission, simple tests of virus shedding would be important to include here.

Reviewer #2:

As a possible additional approach to control FMD Hu and colleagues have produced transgenic pigs that constitutively expressed FMDV specific siRNA. They demonstrated that FMDV infection of fibroblasts derived from these pigs resulted in reduced viral RNA synthesis as compared to fibroblasts from wild-type pigs. They then infected transgenic pigs with FMDV and found that these pigs had much reduced viral RNA synthesis and developed significantly reduced clinical signs as compared to wild-type pigs. As far as I am aware this is the first demonstration of the production of transgenic animals resistant to FMDV using animals naturally susceptible to FMD.

Comments:

1) Since all transgenic pigs developed some clinical signs of FMDV I suggest that the title should be changed to reflect this, e.g.: “Transgenic shRNA pigs have reduced susceptibility to foot and mouth disease virus infection”. Perhaps if the authors use 2 shRNAs to produce transgenic pigs the replication of FMDV could be almost completely blocked.

2) At the end of the third paragraph of the Main text, the authors state that they presumed that FMDV specific shRNA is expressed in various tissues of transgenic pigs (data not shown). It would be valuable to show expression of the shRNA particularly in tissues susceptible to FMDV such as the lungs, skin, etc.

3) It would have been useful to assay for viremia in addition to viral RNA. Hopefully the authors saved serum to perform this assay.

4) Since a concern after infection with a highly infectious virus such as FMDV is the transmission of virus to naïve animals it would have been worthwhile to at least examine the viral RNA load and/or virus titer in nasal secretions. This is an indication of virus transmission by aerosol.

5) The authors indicated that they selected conserved sequences of VP1 from serotype A, O and Asia 1 by gene alignment. How conserved are these sequences across the 3 serotypes? How many strains of each serotype were compared?

6) In their previous manuscript (29) the authors produced transgenic mice expressing siRNAs targeted to the conserved regions of viral proteins 3D and 2B. These viral nonstructural proteins are more conserved proteins than the viral structural proteins. Why did the authors target the viral structural proteins in this manuscript?

7) The authors indicate that the TG pigs were infected with FMDV serotype O, but they do not indicate what FMDV serotype(s) were used to infect TG and WT fibroblasts. Presumably they also used serotype O. It would have been informative if the authors had infected TG fibroblasts with the other FMDV serotypes, i.e., types A and Asia 1. The authors should be able to infect TG and WT fibroblasts with other FMDV serotypes.

8. How old were the transgenic pigs at the time of FMDV challenge?

[Editors' note: further revisions were requested prior to acceptance, as described below.]

Thank you for resubmitting your work entitled “Transgenic shRNA pigs reduce susceptibility to foot and mouth disease virus infection” for further consideration at *eLife*. Your revised article has been favorably evaluated by Charles Sawyers (Senior editor), a Reviewing editor, and one reviewer. The manuscript has been improved but there are some remaining issues that need to be addressed before acceptance, as outlined below:

The additional required revisions are detailed below by our reviewer. These mostly seem straightforward to make. It seems to him important to add lesion scores for all 5 control animals if available; if not, some comment would have to be made to justify the absence. The issue of housing of the animals should be clarified, and there might need to be some change in the comments as to whether the non-TG animals were infected or not (though no problem with the fact that they were protected). It seems desirable to retain Figure 4 from the original version. Finally there is a set of small points that should be readily addressed. The manuscript seems likely to move forward after these issues are dealt with.

Reviewer #2:

Hu and colleagues have addressed most of my concerns. Of course it's unfortunate that there was no serum available to assay for viremia or nasal swabs to assay for virus transmission. However, I still have a few suggestions. In Figure 4 and Figure 4—figure supplement 1 the data for the 5 non-TG pigs is presented as if representing 1 animal. The authors especially need to show the lesion scores for all 5 non-TG animals in Figure 4. This is important to demonstrate that the challenge was severe for all the non-TG animals.

A second extremely important point is regarding the authors response to my comment #3 (Reviewer #2). The authors suggest that the reason there is viral RNA in the serum of both TG and non-TG pigs at all times postinfection is probably due to contamination by aerosol transmission of virus. In the Materials and methods section the authors don't indicate if the TG and non-TG animals were kept in separate rooms. From their answer to this comment it would appear that all animals were kept together in one room. The authors need to clarify this. If the two groups were kept in separate rooms it is difficult to conclude that the presence of viral RNA in the serum of TG animals was due to aerosol contamination and this strongly suggests that the TG animals did in fact get infected. Certainly the TG animals are more protected from disease than the non-TG animals, e.g., see the clinical scores in Figure 4. I believe that it is important to show the data from the first manuscript (Figure 4) that indicates the level of viral RNA in the serum of TG vs non-TG animals over the course of the challenge. It clearly shows much higher levels of viral RNA in the non-TG pigs as compared to the TG pigs. This is in support of the clinical score data. Therefore, I would not eliminate this panel as the authors have done. Clearly this is a very important point. However, I would definitely suggest that in future experiments the control and experimental groups be kept in separate rooms.

Additional data files and statistical comments:

As I stated in my comments in the General assessment section I believe that the authors should include Figure 4 from the original manuscript in this revised manuscript.

---

## [Author Response]

Reviewer #1:

1) How many copies of the transgene are present in each of the eight lines? What tissues (beyond fibroblasts) express the transgene, and at what levels? Is it not surprising that both the high-expressing and low-expressing lines show substantial blocks to virus replication?

We have added data about transgene copies in each of the eight lines. The new data is presented in Figure 2 and additional text describing this experiment is provided (in the third paragraph of the Main text).

Because we needed to keep some TG pigs alive for FMDV infection, only TG 69 and 101 were slaughtered for analysis of the siRNA expression in various organs. Expression of siRNA was detected in all organs tested, including heart, lung, spleen, liver, kidney and muscle, although the siRNA levels were diverse among different tissues (Figure 2).

Yes, it is surprising that both the high-siRNA TG and low-siRNA TG pigs show substantial blocks to virus replication. However, we note that similar results have been reported in prior studies ([23]; Daniel-Carlier et al., 2013). Lyall et al. reported that the siRNA expression levels were unstable and low (below the limit of detection of Northern blot analysis), but transgenic RNAi chickens against influenza prevented virus transmission. The cause of low-expressing lines capable of inhibiting virus infection need be further investigated.

*2) We are given body temperature and vesicle count, but little more about the course of disease (or lack thereof). We have serum levels of virus RNA, but where does virus replicate most extensively (if at all)? Are pathology sections completely free of viral RNA or just indicating lower levels of virus RNA? Is virus shed? (This readout should be very sensitive.) When is it cleared? Does virus persist (for as long as the animals are maintained?) Given the importance of blocking transmission, simple tests of virus shedding would be important to include here*.

We agree that more information about the course of disease is important for the paper. We have added analysis of virus RNA levels in various tissues from all challenged animals at 10th day post-infection (Figure 4). Viral RNA was not detected in heart, lung, spleen, liver, kidney and muscle in the challenged TG pigs, but viral RNA still kept high levels of expression in lymph and lesion in the Non-TG pigs, except heart, lung, spleen and liver (Figure 4). No viral RNA in the Non-TG heart, lung, spleen and liver showed clearance of viral RNA from the tissues, consistent with prior findings (Zhang et al., 2004; [5]). Furthermore, lesion, as a potential source of virus transmission by aerosol, was well known to be the predominant tissue sites of FMDV infection and amplification (Dillon et al., 2011). Levels of viral RNA in foot lesion of TG pigs was much lower than that in the Non-TG pigs (Figure 4), suggesting an encouraged result for blocking transmission.

In the initial manuscript, we showed that levels of virus RNA in serums of both Non-TG and TG pigs increased continuously and immediately after infection until last collecting for serum (tenth day postinfection). However, FMDV generally can be cleared from serum and most of tissues in 7 days postinfection (Zhang et al., 2004; Zhang et al., 2009). We cannot give a reasonable explanation about the results. It is most likely that serums were contaminated by aerosol transmission of virus excreted by those sick animals when blood was collected or any other time. We think the data from contaminated serum is invalid. Thus we remove the data about serum virus RNA from the initial manuscript. We apologize for this.

Reviewer #2:

*1) Since all transgenic pigs developed some clinical signs of FMDV I suggest that the title should be changed to reflect this, e.g.: “Transgenic shRNA pigs have reduced susceptibility to foot and mouth disease virus infection”. Perhaps if the authors use 2 shRNAs to produce transgenic pigs the replication of FMDV could be almost completely blocked*.

We have changed the title to “Transgenic shRNA pigs reduce susceptibility to foot and mouth disease virus infection”.

*2) At the end of the third paragraph of the Main text, the authors state that they presumed that FMDV specific shRNA is expressed in various tissues of transgenic pigs (data not shown). It would be valuable to show expression of the shRNA particularly in tissues susceptible to FMDV such as the lungs, skin, etc*.

We think your comment is important. We have added analysis of the siRNA expression in various organs. Expression of siRNA was detected in all organs tested, including heart, lung, spleen, liver, kidney and muscle, although the siRNA levels were diverse among different tissues. The new data is presented in Figure 2 and additional text describing this experiment is provided (in the third paragraph of the Main text).

*3) It would have been useful to assay for viremia in addition to viral RNA. Hopefully the authors saved serum to perform this assay*.

Because not enough serum was kept, we could not perform this assay. To show more information about challenge experiment, we have added analysis of virus RNA levels in various tissues from all challenged animals at 10th day post-infection (Figure 4). Viral RNA was not detected in heart, lung, spleen, liver, kidney and muscle in the challenged TG pigs, but viral RNA still kept high levels of expression in lymph and lesion in the Non-TG pigs, except heart, lung, spleen and liver (Figure 4). No viral RNA in the Non-TG heart, lung, spleen and liver showed clearance of viral RNA from the tissues, consistent with prior findings (Zhang et al., 2004; [5]).

In the initial manuscript, we showed that levels of virus RNA in serums of both Non-TG and TG pigs increased continuously and immediately after infection until last collecting for serum (tenth day postinfection). However, FMDV generally can be cleared from serum and most of tissues in 7 days postinfection (Zhang et al., 2004; Zhang et al., 2009). We cannot give a reasonable explanation about the results. It is most likely that serums were contaminated by aerosol transmission of virus excreted by those sick animals when blood was collected or any other time. We think the data from contaminated serum is invalid. Thus we remove the data about serum virus RNA from the initial manuscript. We apologize for this.

*4) Since a concern after infection with a highly infectious virus such as FMDV is the transmission of virus to naïve animals it would have been worthwhile to at least examine the viral RNA load and/or virus titer in nasal secretions. This is an indication of virus transmission by aerosol*.

We have added analysis of viral RNA in lesion of the challenged animals. Lesion, as a potential source of virus transmission by aerosol, was well known to be the predominant tissue sites of FMDV infection and amplification (Dillon et al., 2011). Levels of viral RNA in foot lesion of TG pigs was much lower than that in the Non-TG pigs (Figure 4), suggesting an encouraged result for blocking transmission.

5) The authors indicated that they selected conserved sequences of VP1 from serotype A, O and Asia 1 by gene alignment. How conserved are these sequences across the 3 serotypes? How many strains of each serotype were compared?

We apologize for our unclear description in the initial manuscript. Because conserved sequences as siRNA target sites were an alternative strategy preventing escape mutants of virus (Dave et al., 2004), our suppose is to prevent escape mutants of virus by choosing conserved sequences as siRNA target sites in our research. The conserved sequences of VP1 have 95-100% similarity in almost all strains of serotype O of FMDV deposited in GenBank, simultaneously showing 74-100% similarity in 3-5 strains of serotype A and Asia 1. Our aim is not to target serotype A, O and Asia 1 simultaneously. We have added some texts to make it clearer (in the last paragraph of the Main text).

*6) In their previous manuscript (*[29]*) the authors produced transgenic mice expressing siRNAs targeted to the conserved regions of viral proteins 3D and 2B. These viral nonstructural proteins are more conserved proteins than the viral structural proteins. Why did the authors target the viral structural proteins in this manuscript?*

We used the siRNA targeting viral 3D and 2B to produce transgenic pigs; unfortunately cloned pigs could not be obtained efficiently. Many cloned fetus showed developmental abnormalities (mummies or stillborn). The cause has been investigated, and will be published separately.

*7) The authors indicate that the TG pigs were infected with FMDV serotype O, but they do not indicate what FMDV serotype(s) were used to infect TG and WT fibroblasts. Presumably they also used serotype O. It would have been informative if the authors had infected TG fibroblasts with the other FMDV serotypes, i.e., types A and Asia 1. The authors should be able to infect TG and WT fibroblasts with other FMDV serotypes*.

Yes, O serotype of FMDV was used to infect TG and WT fibroblasts. We have indicated it in the revised manuscript. Again we apologize for our unclear description. Because our aim is not to target serotype A, O and Asia 1 simultaneously, and in view of similarity between siRNA and serotype A or Asia 1. We think it is unnecessary to test infection of TG fibroblasts with other FMDV serotypes.

*8*. *How old were the transgenic pigs at the time of FMDV challenge?*

The challenged pigs were 10 to 13 months old. We have specified it in the revised manuscript.

[Editors' note: further revisions were requested prior to acceptance, as described below.]

Reviewer #2:

*Hu and colleagues have addressed most of my concerns. Of course it's unfortunate that there was no serum available to assay for viremia or nasal swabs to assay for virus transmission. However, I still have a few suggestions. In*
Figure 4
*and*
Figure 4—figure supplement 1
*the data for the 5 non-TG pigs is presented as if representing 1 animal. The authors especially need to show the lesion scores for all 5 non-TG animals in*
Figure 4*. This is important to demonstrate that the challenge was severe for all the non-TG animals*.

Thanks for your comments. We have added the data of all 5 Non-TG pigs in Figure 4 and Figure 4—figure supplement 1 in the revised manuscript.

*A second extremely important point is regarding the authors response to my comment #3 (Reviewer #2). The authors suggest that the reason there is viral RNA in the serum of both TG and non-TG pigs at all times postinfection is probably due to contamination by aerosol transmission of virus. In the Materials and methods section the authors don't indicate if the TG and non-TG animals were kept in separate rooms. From their answer to this comment it would appear that all animals were kept together in one room. The authors need to clarify this. If the two groups were kept in separate rooms it is difficult to conclude that the presence of viral RNA in the serum of TG animals was due to aerosol contamination and this strongly suggests that the TG animals did in fact get infected. Certainly the TG animals are more protected from disease than the non-TG animals, e.g., see the clinical scores in*
Figure 4*. I believe that it is important to show the data from the first manuscript (*Figure 4*) that indicates the level of viral RNA in the serum of TG vs non-TG animals over the course of the challenge. It clearly shows much higher levels of viral RNA in the non-TG pigs as compared to the TG pigs. This is in support of the clinical score data. Therefore, I would not eliminate this panel as the authors have done. Clearly this is a very important point. However, I would definitely suggest that in future experiments the control and experimental groups be kept in separate rooms*.

In our viral challenge assay, all animals were housed in one room. We have indicated it in the Materials and methods in the revised manuscript. We have included Figure 4 from the original manuscript in this revised manuscript.

Additional data files and statistical comments:

*As I stated in my comments in the General assessment*
*section I*
*believe that the authors should include*
Figure 4
*from the original manuscript in this revised manuscript*.

We have included Figure 4 from the original manuscript in this revised manuscript.